# Software Language Comprehension using a Program-Derived Semantics Graph

**Roshni G. Iyer    Yizhou Sun    Wei Wang**

University of California, Los Angeles

Los Angeles, CA, 90095, USA

{roshnigiyer, yzsun, weiwang}@cs.ucla.edu

**Justin Gottschlich**

Intel Labs & University of Pennsylvania, USA

Santa Clara, CA, 95054, USA

justin.gottschlich@intel.com

## Abstract

Traditional code transformation structures, such as abstract syntax trees (ASTs), conteXtual flow graphs (XFGs), and more generally, compiler intermediate representations (IRs), may have limitations in extracting higher-order semantics from code. While work has already begun on higher-order semantics lifting (e.g., Aroma's simplified parse tree (SPT), verified lifting's lambda calculi, and Halide's intentional domain specific language (DSL)), research in this area is still immature. To continue to advance this research, we present the *program-derived semantics graph* (PSG), a new graphical structure to capture semantics of code. The PSG is designed to provide a single structure for capturing program semantics at multiple levels of abstraction. The PSG may be in a class of emerging structural representations that cannot be built from a traditional set of predefined rules and instead must be *learned*. In this paper, we describe the PSG and its fundamental structural differences compared to state-of-the-art structures. Although our exploration into the PSG is in its infancy, our early results and architectural analysis indicate it is a promising new research direction to automatically extract program semantics.

## 1   Introduction

*Machine programming* (MP), defined as any system that automates some portion of software, envisions a future where machine learning (ML) can (nearly) automate the entire software development lifecycle [5]. A core open challenge in MP is the ability to automatically extract user intention from code [13]. Exacerbating this problem, new programming languages (PLs) continue to be developed with varying levels of semantic abstraction (e.g., Halide, Python, and SYCL) [7, 10, 12]. Such semantic variability may handicap traditional single dimensional hierarchical structures, such as ASTs, which can generally only represent code at a semantic level for which the syntax exists. Further, these structural limitations might create potential inconsistency and incompatibility in semantic representations from one PL to the next. In this paper, we aim to address this problem with a new structure called the *program-derived semantics graph* (PSG). The PSG's principle purpose is to capture program semantics. However, different from prior structures of which we are aware (e.g., AST, XFG, SPT, etc.) it achieves this in a novel way by introducing a hierarchical structure that varies semantic granularity. The PSG is also graphical in nature which we leverage to identify relationships that might be challenging (or impossible) to represent with tighter constraints such as a tree, which does not allow for certain characteristics like cycles. To capture the richness and nuances of abstract semantic concepts that may be difficult to precisely define by rules, we envision the PSG to pioneer research towards developing *learned* precise programming structural representations. The PSG's representation has numerous applications in software engineering including code translation between PLs, bug detection and root-cause mitigation, code question-answering, and code optimization.

## 2   Related Work

In this section, we discuss recent efforts to extract code semantics and how they compare to our PSG.

**Verified Lifting.**   *Verified lifting* (VL) is a technique that analyzes code from a source PL, lifts its semantics to a higher level representation, then lowers it into a target PL [6]. However, a core

34th Conference on Neural Information Processing Systems (NeurIPS 2020), Computer-Assisted Programming Workshop, Vancouver, Canada. Copyright 2020 by the author(s).

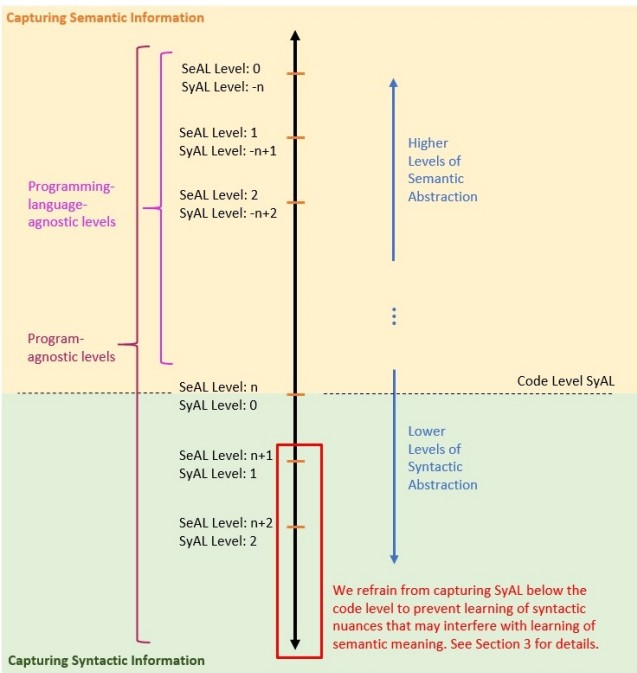

Figure 1: PSG Abstraction Level Spectrum for Semantic Abstraction Levels (SeAL) and Syntactic Abstraction Levels (SyAL), distinguished by color-coding.

challenge with VL may be in its semantic abstraction. Though VL uses a single-level DSL to store semantics, there may be cases where a hierarchical abstraction system is required when moving code between PLs due to their differing abstraction levels (AL). Further, mapping DSL semantic abstractions to hierarchical ALs may improve VL's ability to extract code semantics.

**Neural Code Comprehension.** The goal of the Neural Code Comprehension (NCC) system is to extract code semantics using a fusion of programmatic structural representations in addition to ML-based modeling [2]. NCC introduces a novel structure called the conteXtual flow graph (XFG) to extract semantic meaning through identified data dependencies. The XFG, however, is limited to languages supported by low level virtual machine intermediate representations (LLVM-IR) for which it is lowered to. Moreover, as the XFG is principally grounded to lower-level syntactic representations, its structure may have challenges in extracting the underlying semantic meaning of code.

**Aroma.** Aroma is a novel code recommendation system [9] that emphasizes learning semantics, rather than syntax, through a code's structure using an SPT. We structurally analyze the PSG and Aroma's SPT, which indicates that the PSG may learn more semantics.

## 3 Program-Derived Semantics Graph (PSG) & Language (PSL)

### 3.1 Program-Derived Semantics Graph

The PSG is a multi-tiered representation of program semantics derived from a program's source code. The PSG is PL-agnostic and is a graphical representation of our hierarchical abstract semantic concept language, the PSL. We have designed the PSL in hopes of it being both adaptable and extensible. The Appendix summarizes our motivation for representing the PSG as a graph data structure.

*The PSG Structure.* The PSG incorporates both semantic and syntactic information through hierarchical ALs. [1] As illustrated in Figure 1, each PSG level provides a varying degree of granularity. Higher levels of abstraction capture more abstract and general semantic information, while lower levels of abstraction encode more syntactic and precise information. Providing the correct level of representation of code syntax for semantic analysis is difficult. It continues to be an open challenge

---

[1]Appendix Figure 9 provides a detailed graphical representation of our base PSL, known as base PSG. We adopt the term *base PSG* because it uses a first-order approximation PSL, base PSL. For future work, we aim to provide a more comprehensive construction of the base PSL, by mining previously unseen code structures.

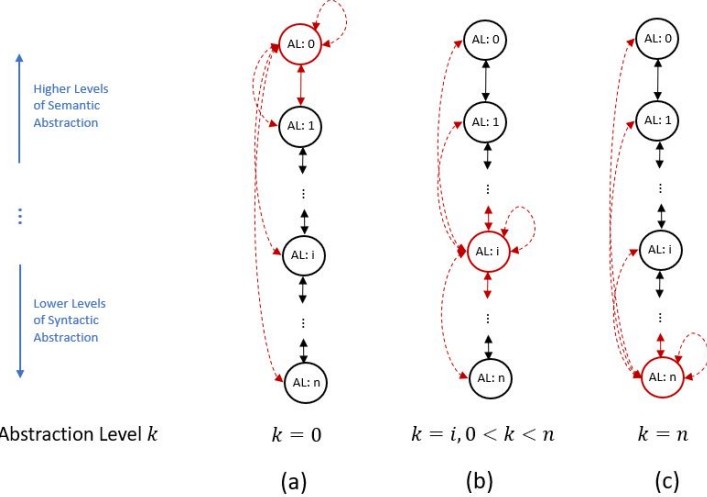

Figure 2: PSG AL Dependencies with potential dependencies (dotted red arrows), and minimum dependencies (solid red arrows); (a) shows dependencies for the highest AL, $k = 0$. (b) shows dependencies for intermediate ALs, $0 < k < n$. (c) shows dependencies for the lowest AL, $k = n$.

in compilers [1], which often perform too much syntactic analysis that they can obfuscate a system's ability to extract semantic meaning. In the Appendix, we describe an example to motivate the need for finding a balance between syntactic and semantic analysis. Our current embodiment of the PSG only incorporates one layer of concrete syntactic abstraction in *SyAL Level: 0*, which is PL-specific. Like Aroma before us, we made this decision to avoid capturing too many nuances of PL syntax, which may interfere with the goal of the PSG to capture program meaning, not implementation details. At the same time, our aim with *SyAL Level: 0* is to help address the problem of learning syntactic information to extract out semantic meaning through categorical classification.

**PSG is Adaptable and Extensible.**    As we are in the early stages of exploring the PSG, it may be unwise to provide a static structure for the PSG. Moreover, as PLs evolve and new PLs are created, our aim is for the PSG to automatically and seamlessly handle such dynamic changes. To address these challenges, we have designed the PSG to be *adaptable* and *extensible*. In this way, we believe the PSG may also likely be the first concrete solution to automatically enhance software even as PLs evolve. The PSG is adaptable in that it is PL-agnostic, and further, as PLs evolve, the PSG is designed to evolve with them with its support of the syntactic layer. The PSG is also extensible since its layers of abstraction are not fixed. This is to accommodate the potential need for adding more layers in order to capture an evolving range of algorithms. The Appendix describes how large-scale software data may be a key component to comprehensively and automatically maitain the PSL and PSG.

### 3.2   Structural Analysis

Due to the PSG's graphical nature and base PSL's design, we believe the PSG may be a scalable structure to represent code bodies that have been historically limited [9], for the following reasons: First, the PSG contains a single AL for capturing syntactic information, with all other levels capturing semantic concept information. Moreover, our base PSL does not include details such as variable or function names. Eliding such information has the byproduct of reducing an instantiation of the PSG's overall size. Second, as the levels of the PSG are raised, the size of ALs shrink. This is because lower levels of abstraction expand upon higher order AL categories by enumerating all objects belonging to those categories. As such, we would expect for the lowest AL, which encodes for syntactic abstraction at the code-level, to be the largest level of abstraction in the PSG. The Appendix provides a conceptual illustration. While prior work that has extracted semantics solely from syntactic information has usually been restricted to inputs of a dozen lines of code or less, we believe it may be possible using the PSG's scalable approach to increase the input sizes.

As a demonstration of the PSG's structure, we show an example of how it captures semantics from two code snippets that are semantically equivalent but syntactically different. We consider two possible implementations of exponentiation (i.e., $x^y$) in C++. One performs the operation recursively,

and the other performs it iteratively. The reader is referred to the Appendix for implementation details. We generate the PSG for both implementations, shown in the Appendix, which we refer to as PSG-recursive and PSG-iterative respectively. We also generate Aroma's SPT of each implementation, shown in the Appendix, which we refer to as SPT-recursive and SPT-iterative respectively.

We perform a rudimentary analysis of the resulting PSGs and SPTs, measuring their corresponding node overlap percentages and differences using the method described below. [2] For this analysis, we refer to the multiset of nodes in a given structure (in this case, either a PSG or SPT) as $N$. For the purposes of this analysis, there are two multisets to consider, which we refer to as $N_1$ and $N_2$. The below analysis is performed twice: once for the resulting PSGs and then again for the resulting SPTs.

1. Calculate multiset intersections: $I_1 = N_1 \cap N_2$ and $I_2 = N_2 \cap N_1$. [3]
2. Calculate percentages of intersection: $P_1 = (|I_1| \div |N_1|)$ and $P_2 = (|I_2| \div |N_2|)$.
3. Calculate percentages distance (degree of difference): $\eta = |P_1 - P_2|$
4. Calculate approximate lower bound: $L = |min(P_1, P_2) - \eta|$
5. Calculate range, $R$, and average, $A$, of structural similarity:
   $R = [L, min(P_1, P_2)]$, $A = (L + min(P_1, P_2)) \div 2$.

Shown above, we devise two novel calculations intended for code similarity. Those are the calculation of (3) the percentages distance, $\eta$, and (4) an approximate lower bound, $L$, that are explained below.

In our analysis of code similarity systems, we note that one code snippet, $S_1$, may have a large percentage intersection with another code snippet, $S_2$. Yet, $S_2$'s intersection with $S_1$ may be small. This irregularity difference is notable because it implicitly argues two opposing views: *(i)* the code snippets are similar and *(ii)* the code snippets are dissimilar. Logically, both views cannot simultaneously hold. To capture these differences, we introduce $\eta$ which grows the greater the difference between the two percentages of intersection. We then introduce $L$, an approximate lower bound, as a penalty for such a difference. The intuition behind $L$ is that similarity overlaps that are relatively small percentage differences (i.e., a small $\eta$) should be penalized less than similarity overlap percentages with relatively large percentage differences (i.e., a large $\eta$). We find that fusing these two calculations presents one possible analysis of both the potential similarity between two code snippets, and of potential limitations of a structural representation used to extract semantic meaning.

Next, we compute the semantic structural similarity from the above procedure between SPT-recursive and SPT-iterative, and between PSG-recursive and PSG-iterative. The computation details are described in the Appendix. For the SPT, we find that $R = [63.70\%, 64.71\%]$ and $A = 64.21\%$. For the PSG, we find that $R = [69.91\%, 70.37\%]$ and $A = 70.14\%$. From this analysis, the PSG defines these code representations to be on average $5.93\%$ more structurally similar compared to the SPT. Although these results are anecdotal, we believe they provide early intuition on how the PSG may be used and how it generally compares, structurally, to Aroma's SPT. Construction of the PSG and our measurement of semantics does not study a program by its execution or use program synthesis techniques such as code input and output analysis. Rather, our technique constructs and analyzes a PSG without requiring code compilation, a practical assumption in many cases for code development.

## 4   Conclusion and Future Work

This position paper provides a conceptual framework for reasoning about code semantics through a program-derived semantics graph (PSG). In it, we presented the concept and intuition of both the PSG and the program-derived semantics language (PSL). The PSG is a graphical representation of our hierarchical abstract semantic concept language, the PSL, which we have designed in the hopes of it being both adaptable and extensible. We described their fundamental structure and intuition, and illustrated both the PSG and PSL against Aroma's state-of-the-art simplified parse tree.

For future work, we plan to analyze the PSG and other state-of-the-art systems, like Aroma, against larger code corpora. We also leave the problem of automatically generating the PSG from code as part of future work that is currently work-in-progress. While we have early ideas for tackling this problem, the system design is outside the scope of this paper.

---

[2]Though not intended to be comprehensive, our analysis is one approach for computing semantic similarity.

[3]$N$ represents a multiset. As such, the intersection of $N_1 \cap N_2$ is not necessarily equivalent to $N_2 \cap N_1$.

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

# Appendix

## A   Motivation for a Graphical Representation

We were motivated to represent the PSG as a graph data structure for some of the following reasons:

1. Graphs can effectively encode structural information, or preserve syntactic meaning, through parent-child-sibling node hierarchy. While both graphs and trees can preserve hierarchical structural information, graphs are more general. This generality may be useful when working on an open research question, like code similarity, where added flexibility may result in a broader exploration of solutions.

2. Graphs can be effective representations for graph neural networks (GNNs) used to learn latent features or semantic information. Relational graph convolutional networks (R-GCNs) [11] are a class of GNNs that apply graph convolutions on highly multi-relational graphs, like the PSG, to learn graph structure and semantic meaning. Models using GNN-based approaches have achieved promising results in the domain of representation learning [8].

3. The semantics of certain software abstraction levels may be more easily represented using a graph. One concrete example of this is illustrated in Neural Code Comprehension [2]. As shown in their contextual flow graph, dependencies of data and control flow may take on a graph structure, where two nodes can be connected by more than one edge. The authors show that a tree structure would be insufficient for capturing such (potentially) cyclic dependencies.

## B   Example: The delicate balance between syntax and semantic analysis

Consider a program that seeks to manipulate strings. In many languages, one could store such information in many ways. For example, the following are some ways one could implement a string in C++: `std::string`, `char[]`, `wchar_t[]`, `std::vector<char>`, `std::array<wchar_t, size>`, to name a few. If a semantic extraction system focuses too deeply on such implementation nuance, it may learn that two programs, which are identical in terms of semantics, are dissimilar due to divergent implementation details. On the other hand, ignoring all such details may eliminate information that is critical in interpreting the semantics of the program. For example, capturing the semantic details, through syntactic interpretation that a single variable is being used to manipulate information, rather than a collection of variables, could be critical in understanding cardinality constraints of a particular problem [3].

It is for these reasons that the PSG captures some syntactic information to tailor semantic learning to the PL-specific functionalities as determined by the syntax. The last abstraction level of the PSG is PL-specific. Our aim with this layer is to help to address the problem of learning syntactic information to extract out semantic meaning through categorical classification.

## C   Structural Analysis of the PSG

As shown in Figure 3, when the levels of the PSG are raised, the size of abstraction level $i$, denoted as $m_i$, shrinks. This is because lower levels of abstraction expand upon higher order abstraction level categories by enumerating all objects belonging to those categories. As such, we would expect for the lowest abstraction level, which encodes for syntactic abstraction at the code-level, to be the largest level of abstraction in the PSG.

As a demonstration of the PSG's structure, we show an example of how it captures semantics from two code snippets that are semantically equivalent but syntactically different. Further, using this example, we compare the PSG to Aroma's SPT. We consider two possible implementations of exponentiation (i.e., $x^y$) in C++, shown in Figure 4. One performs the operation recursively, and the other performs it iteratively. We generate the PSG of both implementations shown in Figures 5 and 6. We refer to these representations as PSG-recursive and PSG-iterative. We also generate Aroma's SPT of each implementation in Figures 7 and 8. We refer to these representations as SPT-recursive and SPT-iterative.

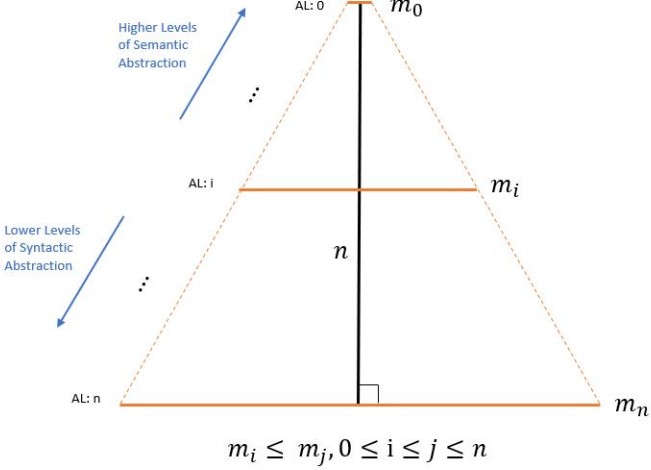

$$m_i \leq m_j, 0 \leq i \leq j \leq n$$

Figure 3: Relationship between PSG Abstraction Level (AL) size and count. There are $n$ ALs where the size of each AL $i$ is $m_i$ which indicates the number of semantic concepts AL $i$ captures.

**Implementation 1**

```
0 signed int recursive_power(signed int x, unsigned int y)
1 {
2    if (y == 0)
3        return 1;
4    else if (y % 2 == 0)
5        return recursive_power(x, y / 2) *
            recursive_power(x, y / 2);
6    else
7        return x * recursive_power(x, y / 2) *
            recursive_power(x, y / 2);
8 }
```

**Implementation 2**

```
0 signed int iterative_power(signed int x, unsigned int y)
1 {
2    signed int val = 1;
3    while (y > 0) {
4        val *= x;
5        y -= 1;
6    }
7    return val;
8 }
```

Figure 4: Two functions computing exponentiation (i.e., $x^y, x \in \mathbb{R}$ and $y \geq 0$) in C++. Implementation 1 is recursive. Implementation 2 is iterative. The functions are semantically equivalent but syntactically inequivalent.

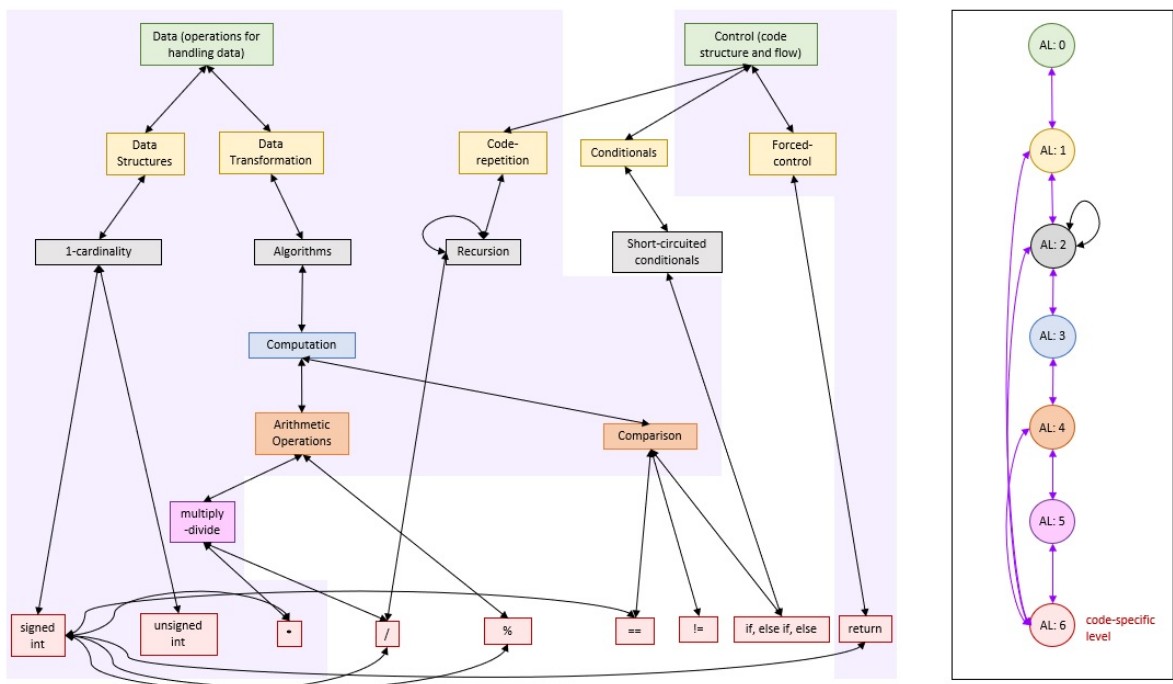

Figure 5: PSG of Recursive Power Function. The shaded region denotes overlap in the nodes of the PSG for the iterative power function shown in Figure 6. These total 17 of the 24 total nodes, a 70.83% overlap.

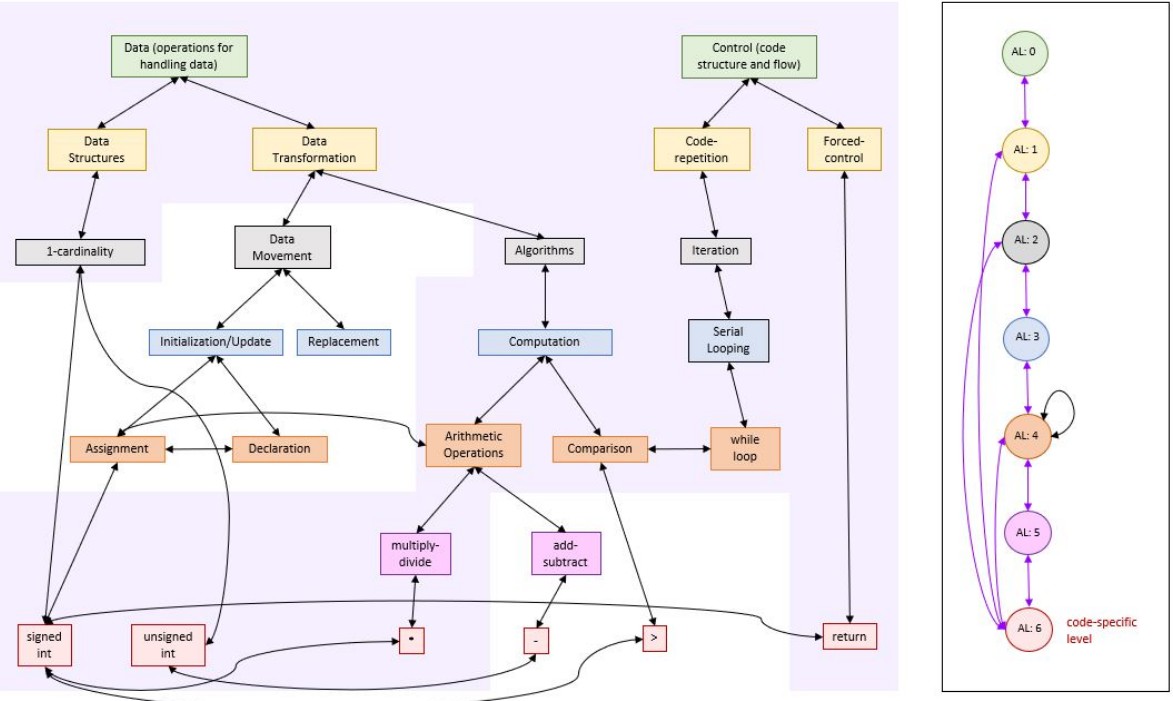

Figure 6: PSG of Iterative Power Function. The shaded region denotes overlap in the nodes of the PSG for the recursive power function shown in Figure 5. These total 19 of the 27 total nodes, a 70.37% overlap.

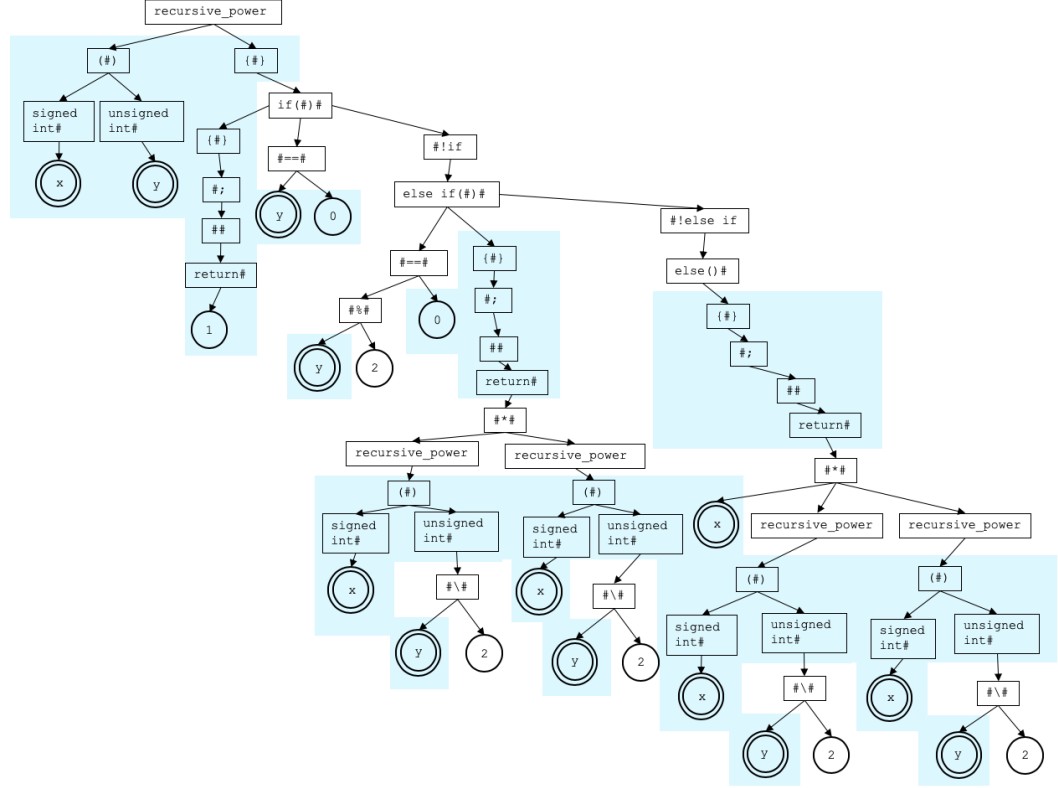

Figure 7: SPT of Recursive Power Function. The shaded region denotes overlap in the nodes of the SPT for the iterative power function shown in Figure 8. These total 44 of the 68 nodes, a 64.71% overlap.

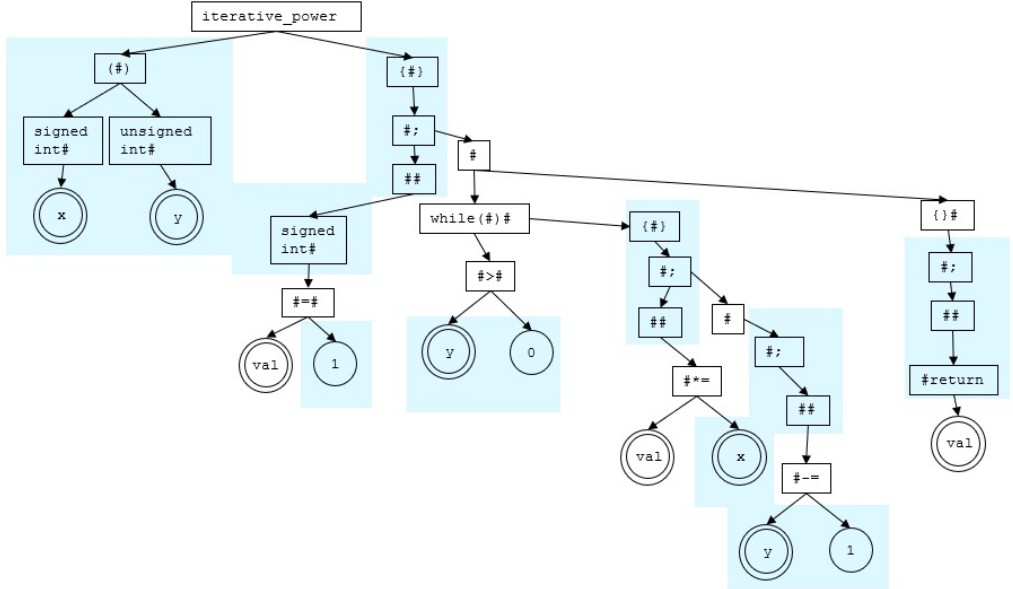

Figure 8: SPT of Iterative Power Function. The shaded region denotes overlap in the nodes of the SPT for the recursive power function shown in Figure 7. These total 23 of the 35 nodes, a 65.71% overlap.

# D   Structural Analysis Computation: SPT and PSG

For SPT:

1. $|N_1| = 68$, $|N_2| = 35$, $|I_1| = 44$, and $|I_2| = 23$

2. $P_1 = (|I_1| \div |N_1|) = \frac{44}{68}$, $P_2 = (|I_2| \div |N_2|) = \frac{23}{35}$

3. $\eta = |P_1 - P_2| = \frac{6}{595}$

4. $L = |min(P_1, P_2) - \eta| = \frac{379}{595}$

5. $R = [L, min(P_1, P_2)]$ $\boxed{= [63.70\%, 64.71\%]}$
   $A = (L + min(P_1, P_2)) \div 2$ $\boxed{= 64.21\%}$

For PSG:

1. $|N_1| = 24$, $|N_2| = 27$, $|I_1| = 17$, and $|I_2| = 19$

2. $P_1 = (|I_1| \div |N_1|) = \frac{17}{24}$, $P_2 = (|I_2| \div |N_2|) = \frac{19}{27}$

3. $\eta = |P_1 - P_2| = \frac{1}{216}$

4. $L = |min(P_1, P_2) - \eta| = \frac{151}{216}$

5. $R = [L, min(P_1, P_2)]$ $\boxed{= [69.91\%, 70.37\%]}$
   $A = (L + min(P_1, P_2)) \div 2$ $\boxed{= 70.14\%}$

## D.1   Learning the PSL With Data

Data may be a key component to enable a comprehensive and automatically maintained PSL. While base PSG is a working graphical representation of the first-order approximation PSL, it is non-exhaustive of all semantic concepts and their dependencies. To mitigate this weakness, we believe it may be possible to augment the base PSL through a continuously refined learning system, which will aim to learn new semantic concepts and dependencies of PLs from data patterns (i.e., anomalies) it has not previously observed. With the emergence of publicly large available code repositories (e.g., as of this writing, GitHub has around 200 million repositories [4]), and the growing magnitude of the web itself, we believe an automated and synthesized comprehensive PSL may be within our technological reach.

# E Base PSG Abstraction Levels

AL: 0

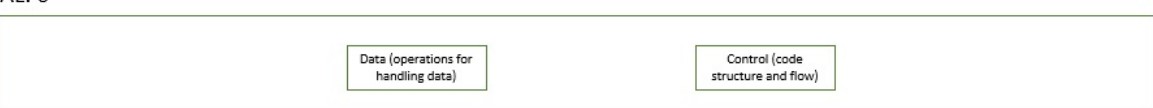

AL: 1

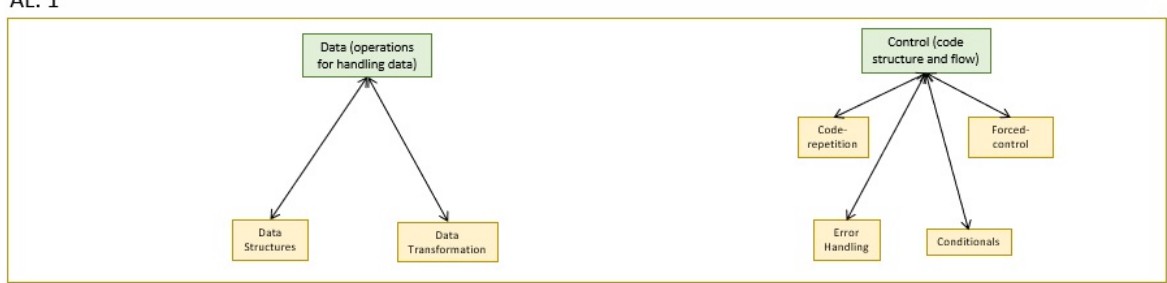

AL: 2

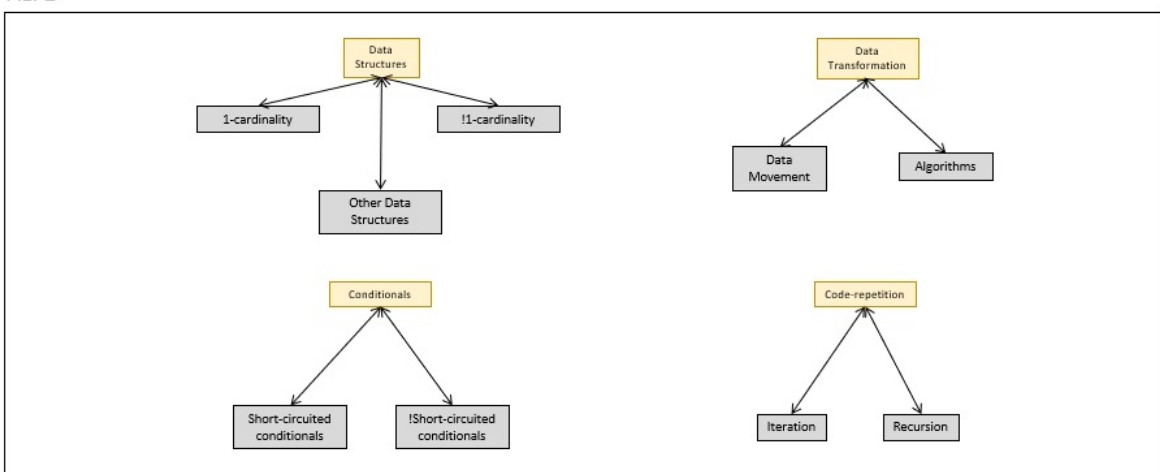

AL: 3

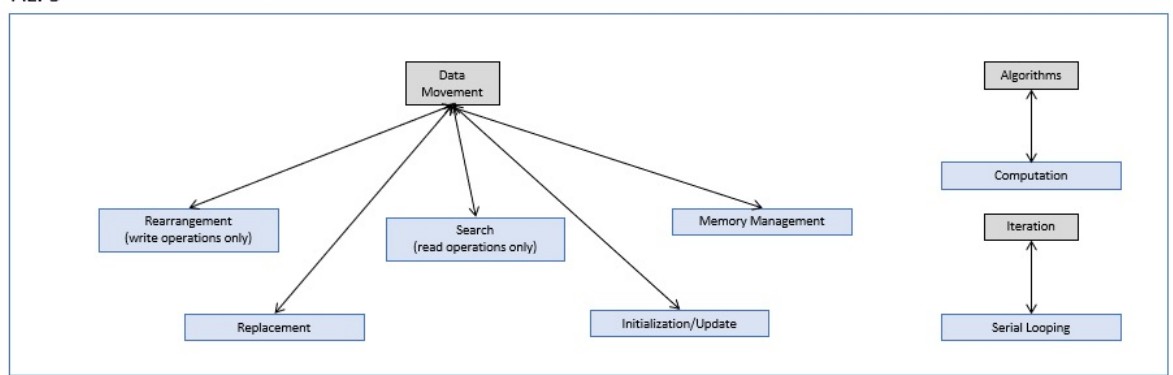

AL: 4

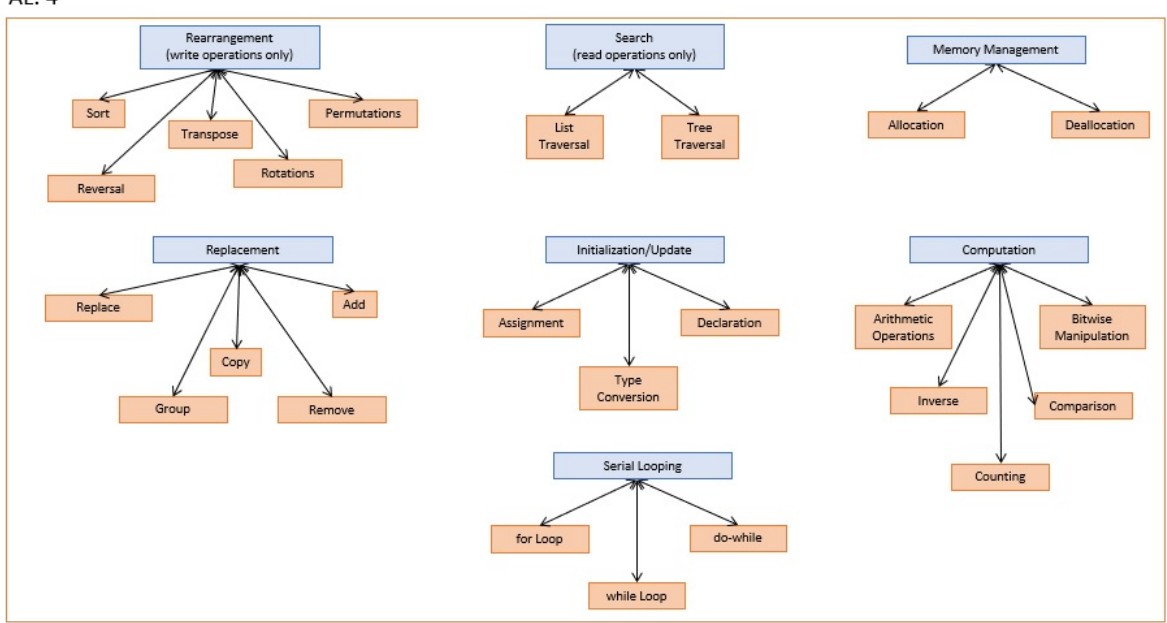

AL: 5

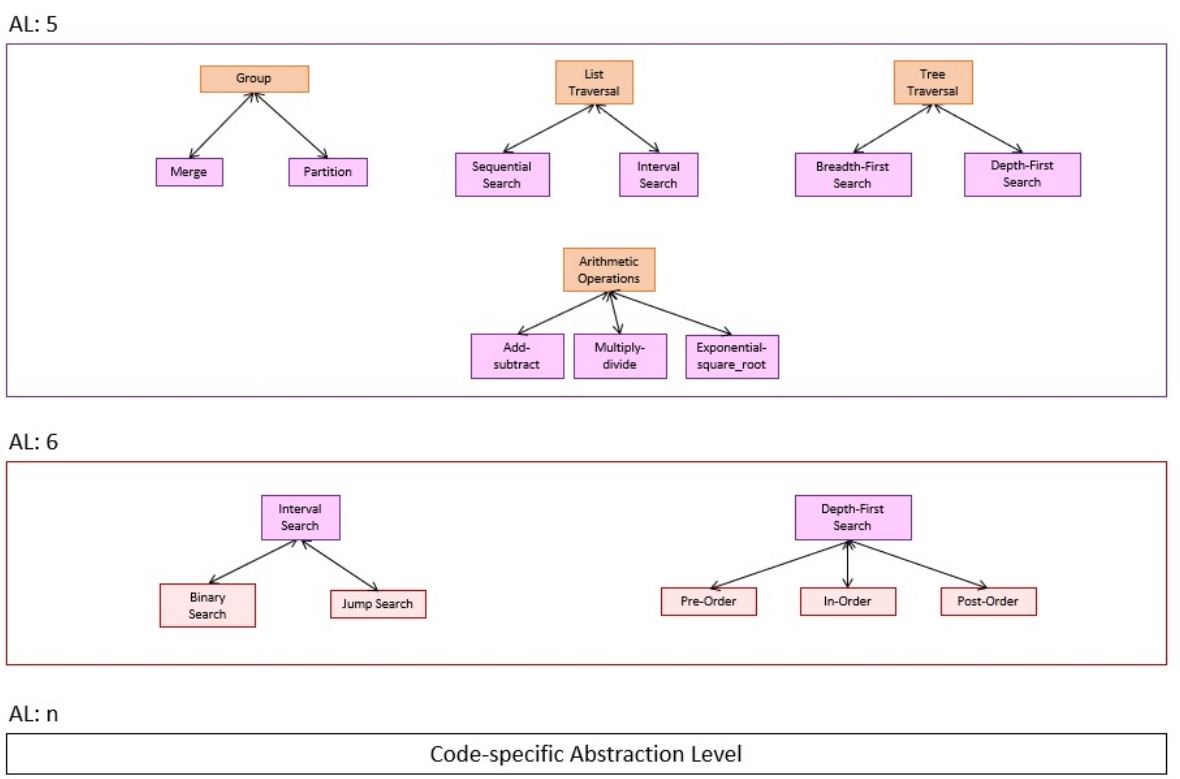

AL: 6

AL: n

Code-specific Abstraction Level

Figure 9: Detailed Abstraction Level (AL) construction of base PSG

