# OpenReview forum: "Software Language Comprehension using a Program-Derived Semantics Graph"
_NeurIPS.cc/2020/Workshop/CAP — NeurIPS 2020 CAP Workshop_

### Official Review · AnonReviewer1 · 2020-10-22
**We wish to create a hierarchy of abstractions to represent code (similar to LLVM but better), such that higher level concepts can be represented**

**Rating:** 6
**Confidence:** 3

**Review:**

Such representation would indeed be useful. As most of research working over traditional programs (C, python) – as opposed to a small synthetic DSL – for the most part reason over the syntax of these programs. By having a better representation that abstract away the syntactical differences we can learn over these programs better.

A key question to answer for any representation would be, what is "closeness". This work propose such measurement by hand-crafting PSG of two programs, and measure some notion of graph similarity. This is a step in the right direction.

One useful measurement of semantics of a piece of program is execution. Have you thought about just run the code? If they give same output on the same input, clearly semantically speaking, on the highest level, they match. You can then try to look at the intermediate values of execution, and try to do some similarity there too, to get a lower level similarity. For instance "x + 2" and " 2(x+1) - x" are semantically equivalent.

---

### Decision · Program_Chairs · 2020-11-02

**Decision:**

Accept

**Comment:**

As the review is overall positive, I recommend acceptance.